# Metal Complexes of Schiff Bases Prepared from Quinoline-3-Carbohydrazide with 2-Nitrobenzaldehyde, 2-Chlorobenzaldehyde and 2,4-Dihydroxybenzaldehyde: Structure and Biological Activity

Mahmoud Sunjuk [1,*], Lana Al-Najjar [1], Majed Shtaiwi [1], Bassam I. El-Eswed [2], Kamal Sweidan [3], Paul V. Bernhardt [4], Hiba Zalloum [5] and Luay Al-Essa [6]

[1]  Department of Chemistry, Faculty of Science, The Hashemite University, Zarqa 13133, Jordan; lanam_s@hu.edu.jo (L.A.-N.); majedh@hu.edu.jo (M.S.)
[2]  Department of Basic Sciences, Zarqa College, Al-Balqa Applied University, Al Salt 19117, Jordan; bassameswed@bau.edu.jo
[3]  Department of Chemistry, The University of Jordan, Amman 11942, Jordan; k.sweidan@ju.edu.jo
[4]  School of Chemistry and Molecular Biosciences, University of Queensland, Brisbane 4072, Australia; p.bernhardt@uq.edu.au
[5]  Hamdi Mango Center for Scientific Research (HMCSR), The University of Jordan, Amman 11942, Jordan; hmzalloum@ju.edu.jo
[6]  Department of Pharmacy, Faculty of Pharmacy, Al-Zaytoonah University of Jordan, Amman 11733, Jordan; luayyousif@yahoo.com
*   Correspondence: mahmoud.sunjuk@hu.edu.jo

**Abstract:** Three Schiff base ligands, NQ, CQ and HQ, were prepared from the reaction of quinoline-3-carbohydrazide with 2-nitrobenzaldehyde, 2-chlorobenzaldehyde and 2,4-dihydroxybenzaldehyde, respectively, and were investigated for their coordination to Cu (II), Ni(II), Co(II), Cd(II), Cr(III) and Fe(III) chlorides. The NQ preparation and the X-ray structure of NQ and CQ, as well as the transition metal complexes of NQ, CQ and HQ, were reported for the first time. FTIR, $^1$H-NMR, magnetic susceptibility and elemental analysis were used to study the coordination of ligands to the metal ions. Based on the magnetic susceptibility and elemental analysis results, octahedral structures of the complexes such as [CuL$_2$Cl(OH)], [FeL$_2$Cl$_2$(OH)] and [CoL$_2$Cl(OH)] were proposed for L = NQ, CQ and HQ. The relatively large Cd(II) exhibited [CdL$_3$(OH)$_2$]. The FTIR study revealed that NQ and CQ are coordinated to the metal ions via azomethine nitrogen and carbonyl oxygen while HQ through azomethine nitrogen and phenolic oxygen. Despite the high solvation power of DMSO solvent in $^1$H-NMR experiments, the azomethine HC=N peak at 9.3 ppm is the most affected by complexation with metal ions. On the other hand, quinoline nitrogen seems to be a weaker coordinating site than the azomethine nitrogen. The HQ ligand, containing phenolic groups, and its complexes with Cu and Ni were found to have inhibitory effects on human breast adenocarcinoma MCF-7 and human chronic myelogenous leukemia K562. Nevertheless, metal ions did not exhibit a significant synergistic effect on the antiproliferative activity of the ligands investigated.

**Keywords:** Schiff bases; quinoline-3-carbohydrazide; benzaldehyde; transition metal complexes

## 1. Introduction

Schiff bases (general formula of R$_1$R$_2$C=NR′ (R′ ≠ H)) are a subclass of imines [1]. Condensation of an aliphatic or aromatic amine and carbonyl compound is the routine method for synthesis of Schiff bases [2]. Schiff bases are very common ligands in organometallic chemistry, which coordinate to metal ions through the nitrogen atom of the azomethine [3,4]. Schiff bases found many applications in analytical, biological [3] and pharmaceutical fields. They have antibacterial, antimalarial, anti-inflammatory, antiviral and antipyretic

properties [5]. Furthermore, transition Schiff base complexes with metal ions exhibited antibacterial and antifungal activities [6,7].

Quinolines, some of the partners used in synthesis of Schiff bases prepared in the present work, are useful in different applications in pharmacology since they have anti-inflammatory, analgesic, antibacterial, antifungal, antimalarial, anthelmintic, anticonvulsant and cardiotonic activities [8]. Halogen-substituted quinoline compounds have been of particular interest since the halogen atom has special biological activities as in the case chloroquines [9–11]. There is growing evidence that chloroquine and hydroxychloroquinolinies, which are broadly used as antimalarial and immunomodulatory drugs, can be used in the treatment of patients with COVID-19 infection [12].

Schiff bases of 2-(quinolin-8-yloxy) acetohydrazide and their Cu(II) and Zn(II) metal complexes were investigated for their antimicrobial activity. The Cu complex was the most effective against Gram-positive S. aureus and E. faecalis bacteria. Adsule et al. reported a 1:1 copper complex of Schiff base quinoline-2-carboxaldehyde [CuL$_2$Cl$_2$] exhibiting antiproliferative and proapoptotic activity in PC-3 and LNCaP prostate cancer cells (IC50 of 4 and 3.2 μM, respectively) [13]. Patil and Vibhute reported synthesis of a Schiff base, namely, (*E*)-*N*′-((2-hydroxyquinolin-3-yl) methylene)-4-methylbenzenesulfonohydrazide, and its metal complexes with Cu (II), Ni (II), Co (II) and Cd (II). The metal complexes were found to be more active than the ligand in vitro cytotoxicity with a human lung cancer cell line (A-549) and human breast cancer cell line (MCF-7), among which the Cu complex exhibited the highest activity [14].

In the literature, the azomethine nitrogen is conjugated with phenolic oxygen [14–18], quinoline [13,19–21] or phenolic oxygen and carbonyl [22,23] to facilitate the chelating/coordinating properties of ligands to metal ions. The present work will describe the synthesis of Cu(II), Co(II), Ni(II), Cd(II), Cr(III) and Fe(III) complexes with three Schiff base ligands (NQ, CQ and HQ) which were prepared by condensation of quinoline-3-carbohydrazide with aromatic aldehydes: 2-nitrobenzaldehyde, 2-chlorebenzaldehyde and 2,4-dihydroxybenzaldehyde, respectively (Scheme 1). The ligands have an array of quinolone nitrogen, carbonyl oxygen and azomethine nitrogen, as well as phenolic oxygen (in the case of HQ), rendering potential coordination sites for metal ions. FTIR, NMR spectra, magnetic susceptibility and elemental analysis of the complexes are used to study the coordination nature of ligand to metal. The complexes, as well as the ligands, were investigated for their antiproliferative activity against human breast adenocarcinoma MCF-7, chronic myelogenous leukemia K562 and dermal fibroblast cell lines.

**Scheme 1.** Synthesis of Schiff base ligands.

## 2. Results and Discussion

### 2.1. Synthesis of Schiff Base Ligands

NQ, CQ and HQ ligands were synthetized as shown (Scheme 1) using a modified method of [24], with % yields ranging from 80 to 86%. Noteworthily, NQ was prepared for the first time. The colors of NQ, CQ and HQ were pale yellow. The relatively high melting point of HQ (288–290 dec.), which was much higher than that of NQ (235–237 °C) and CQ (223–225 °C), can be ascribed to the strong H-bonding between the phenolic groups of HQ.

The FTIR bands of NQ, CQ and HQ ligands in comparison with quinoline-3-carbohydrazide are given in Table 1. The FTIR bands of the ligands reflected disappearance of the sharp band of the amine $\nu(NH_2)$ found in the spectrum of quinoline-3-carbohydrazide and the appearance $\nu$ (-HC=N-). The $\nu(NO_2)$ bands in the spectrum of 2-nitrobenzaldehyde were shifted to the lower frequency upon formation of NQ. Furthermore, the $\nu(C\text{-}OH)$ bands in the spectrum of 2,4-dihydroxybenzaldehyde were shifted to the lower frequency in the case of HQ. On the other hand, the $\nu(C\text{-}Cl)$ band in the spectrum of 2-chlorobenzaldehyde was shifted to lower frequency in the case of CQ.

**Table 1.** Comparison of FTIR stretching vibration bands (cm$^{-1}$) of NQ, CQ and HQ ligands with the amine (quinoline-3-carbohydrazide, PQ).

| Code | ν NH2 | ν C=O | ν N=C (Quinoline) | ν -HC=N- (Azomethine) | ν C-NO$_2$ | ν C-Cl | ν -C-OH |
|---|---|---|---|---|---|---|---|
| PQ | 3193 | 1656 | 1617 | | | | |
| NQ | | 1645 | 1599 | 1619 | 1343 | | |
| CQ | | 1641 | 1595 | 1619 | | 741 | |
| HQ | | 1660 | 1608 | 1630 | | | 1493, 1455 |

The $^1$H-NMR (400 MHz, DMSO-d$_6$) spectra of NQ, CQ and HQ ligands (Figure S1) showed two types of singlet peaks. The first was observed at δ = 12.54, 12.43 and 12.23 ppm, respectively, which was assigned to the N-H proton. The second was at δ = 9.36, 9.36 and 9.34 ppm, respectively, assigned to the azomethine proton (-HC=N-). The doublet at 8.95, 8.93 and 8.93 ppm, respectively, was assigned to the quinoline protons 1 and 3. The sets of peaks observed in the range from 6 to 8 ppm were ascribed to the remaining aromatic protons. Two broad singlet peaks at δ = 10.01 ppm and δ = 12.23 ppm were attributed to the two hydroxyl protons of HQ.

$^{13}$C-NMR spectra of NQ, CQ and HQ ligands are shown in Figure S2. The resonances of the quinoline ring (-C=N-) were observed at δ = 149.4, 149.5 and 149.3 ppm, respectively, and the Schiff base azomethine carbon atom (-HC=N-) was observed at δ = 144.13, 144.75 and 149.09 ppm, respectively. The peaks at δ = 162.42, 162.27 and 161.56 ppm, respectively, were assigned to the amidic group (-HN-C=O). Each of the δ = 148.77 and 149.09 ppm peaks were assigned to carbon linked to the nitro group (-C-NO$_2$) and carbon linked to the chloride group (-C-Cl), respectively. The peaks at δ = 161.41 and 159.99 ppm were assigned to the two carbons linked to the hydroxyl groups (-C-OH). The unsubstituted aromatic carbons were observed in the range of 125 to 148 ppm.

### 2.2. X-ray Structures of NQ and CQ

We report here the X-ray structures of CQ and NQ. Interestingly, these represent the first crystal structures of a 3-substituted quinolyl hydrazone in the literature. More common are 2-substituted quinoline analogs [25].

The bond lengths and angles within CQ·H$_2$O (Figure 1) are as expected and the molecule is essentially planar with all dihedral angles within 7° of either 0 or 180°. H-bonding is a feature and the water molecule accepts an H-bond from the hydrazone NH group (N2-H2...O2 2.03 Å, 162.1°) while donating H-bonds to the quinoline N-atom (O2-H2B...N1′ 2.06 Å, 166.6°, symmetry operation $-x + 1$, $-y + 1$, $-z + 1$) and carbonyl O-atom

(O2-H2A...O1″ 2.08 Å, 162.8°, symmetry operation $x − 1/2$, $−y + 3/2$, $−z + 1$) of different hydrazone molecules.

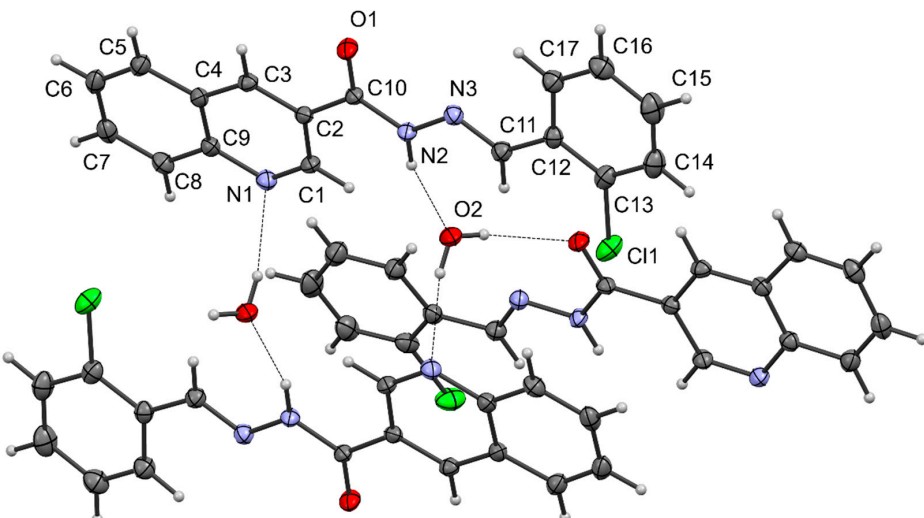

**Figure 1.** Crystal structure of CQ·$H_2O$ showing intermolecular H-bonding (30% probability ellipsoids). Figure produced with Mercury [26].

The crystal structure of NQ·$H_2O$ (Figure 2) is similar and again the water molecule plays a central role in bridging adjacent hydrazone molecules through hydrogen-bonding interactions. The water molecule accepts an H-bond from the hydrazone NH group (N2-H2...O4′ 2.02 Å, 165.4° symmetry operation $−x + 1$, $y + 1/2$, $−z + 3/2$) and donates H-bonds to the quinoline N-atom (O4-H4A...N1″ 2.051 Å, 168.2°, symmetry operation $x$, $−y + 3/2$, $z + 1/2$) and carbonyl O-atom (O4-H4B...O1 2.03 Å, 164.7°). The hydrazone is less planar which is attributable to the nitro group which twists out of the plane of the phenyl ring (C14-C13-N4-O3 31.3(4)°) and also causes the phenyl ring to twist out of the plane of the quinoline rings (N3-C11-C12-C17 16.5(4)°).

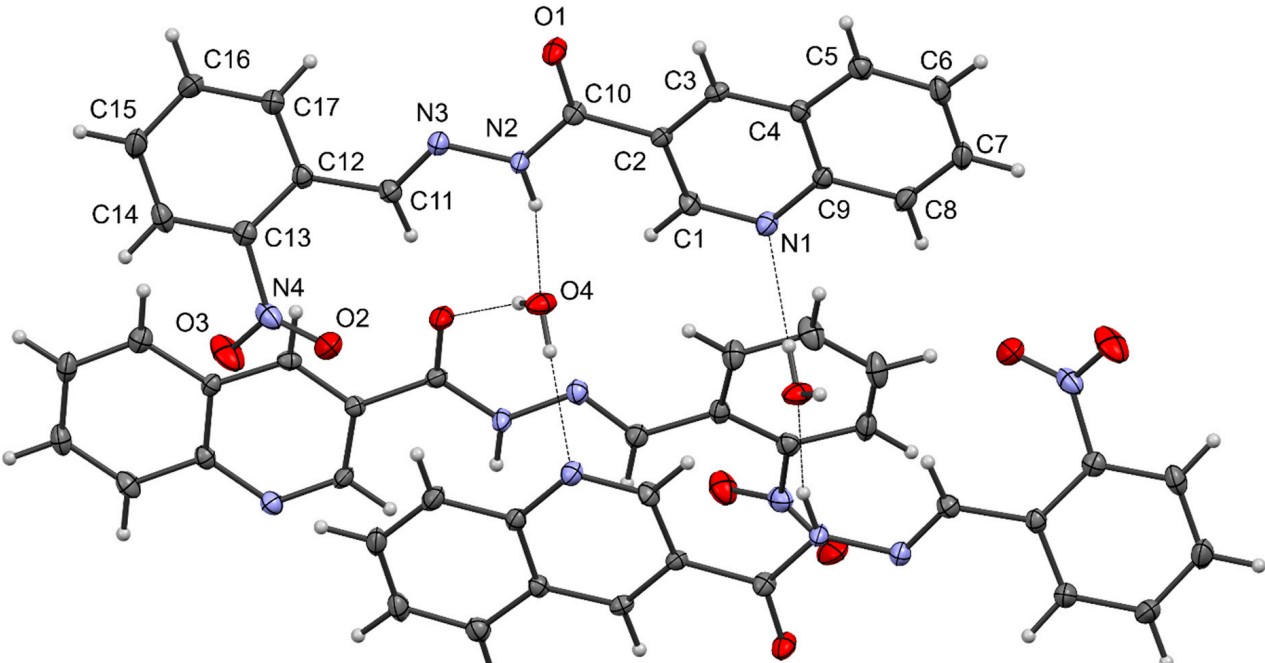

**Figure 2.** Crystal structure of NQ·$H_2O$ showing intermolecular H-bonding (30% probability ellipsoids).

### 2.3. Synthesis of Metal Complexes

The general procedure for synthesis of metal complexes was prepared following a modified procedure of [27,28] (Scheme 2). A methanolic solution of the Schiff base ligand (NQ, CQ and HQ) was refluxed with an ethanolic solution of metal ions (2:1 molar ratio) for 1 h. The solution was evaporated until the complex was precipitated which was collected by suction filtration, washed with cold ethanol and dried at 60 °C in a vacuum oven.

**Scheme 2.** Synthesis of metal complexes.

### 2.4. Physical Properties of Complexes

All the complexes have distinct colors from the free ligands. The melting points of complexes were comparable with those of the ligands (in the range from 200 to 300 °C) except those of Fe-NQ (78–80 °C), Co-NQ (100–102 °C) and Cr-HQ (108–100 °C) which have remarkably low melting points. All metal complexes are weak electrolytes since they are insoluble in water. The complex of Cu(II) with 2-(quinolin-8-yloxy)acetohydrazide Schiff base was found to be of nonionic nature (weak electrolyte) [16]. Furthermore, the Cu(II) and Ni(II) complexes with 2-quinoline carboxaldehyde Schiff base of S-methyldithiocarbazate were of nonelectrolyte nature [19]. The complexes are insoluble in diethyl ether and petroleum ether but soluble in ethanol, dimethyl sulfoxide (DMSO) and N,N-dimethylformamide (DMF).

### 2.5. Magnetic Susceptibility of Metal Complexes

The observed magnetic moments ($\mu$eff, B.M.) of the prepared complexes are given in Table 2. Spin-only magnetic moment can be calculated from $\mu$ s.o. = $\sqrt{(4S(S+1))}$ B.M.; where S is the spin quantum number of unpaired electrons. The measured magnetic moments of Cu-NQ, Cu-CQ and Cu-HQ complexes were 2.06, 2.17 and 1.96 B.M., respectively, which are in agreement with the reported experimental range for Cu(II) octahedral complexes (1.7–2.2 B.M.) [28,29]. A magnetic moment of 1.74 was reported for the Cu(II) complex with an analogous Schiff base (*E*)-*N*'-((2-hydroxyquinolin-3-yl) methylene)-4-methylbenzenesulfonohydrazide [14]. Since the observed values are higher than $\mu$ s.o. ($d^9$, S=1/2) = $\sqrt{(4(1/2) (1/2+1))}$ = 1.73 B.M., orbital contribution of the magnetic moment is possible. Similarly, the observed values of Cr-NQ, Cr-CQ and Cr-HQ complexes (3.93, 3.77 and 3.75 B.M., respectively) were within the reported range for Cr(III) octahedral complexes which is 3.7–3.9 [28] and $\mu$ s.o. (d3, S=3/2) = $\sqrt{(4(3/2) (3/2+1))}$ = 3.87. Furthermore, the observed values of Fe-NQ and Fe-HQ complexes (5.92 and 5.99 B.M., respectively) were within the reported range for high spin Fe(III) octahedral complexes which is 5.7–6.0 [28] and $\mu$ s.o. (d5, S=5/2) = $\sqrt{(4(5/2) (5/2+1))}$ = 5.92. The low magnetic moment of Fe-CQ (2.55) may indicate decomposition. Since the measured magnetic moments of Cu(II), Cr(III) and Fe(III) complexes were close to $\mu$ s.o., then the orbital contribution to magnetic moment is negligible due to the symmetrical occupation of the octahedral electronic configuration of $t_2g$ [30,31]. Tetrahedral geometry for Cu(II) and Cr(III) complexes is not possible since

unsymmetrical $t_2$ occupation of electrons in tetrahedra Cu(II) and Cr(III) ($e^4 t_2{}^5$ and $e^2 t_2{}^1$, respectively) results in orbital contribution to magnetic moment [30].

**Table 2.** Magnetic susceptibility (Bohr magnetons, B.M.) of NQ, CQ and HQ complexes.

| Ligand | NQ | CQ | HQ |
|---|---|---|---|
| Cu | 2.06 | 2.17 | 1.96 |
| Cd | Dia | −0.14 | −0.18 |
| Cr | 3.93 | 3.77 | 3.75 |
| Fe | 5.92 | 2.55 | 5.99 |
| Co | 2.77 | 5.03 | 1.94 |
| Ni | 3.51 | 2.96 | 2.95 |

The magnetic moments of Ni-NQ, Ni-CQ and Ni-HQ complexes (3.51, 2.96 and 2.95 B.M., respectively) were close to those of $\mu$ s.o. ($d^8$, S=1) = $\sqrt{(4(1)(1+1))}$ = 2.83 B.M. and those reported for Ni(II) octahedral complexes (2.8–3.5) [28]. Tetrahedral geometry, which was reported to have magnetic moment in the range 4.2–4.8 [28], can be excluded. Square planar geometry for Ni(II) complexes is excluded also since it is diamagnetic [32,33].

The magnetic moments of Co-NQ, Co-CQ and Co-HQ complexes are 2.77, 5.03 and 1.9 B.M., respectively, among which only the value of Co-CQ is close to those reported for Co(II) high-spin octahedral complexes (4.3–5.2) [28,34]. The magnetic moments of Co-NQ and Co-HQ are close to those of low-spin cobalt(II) complexes which were reported in the range of 1.9 to 2.8 B.M. [35]. A values of magnetic moments of 1.73, 3.25 and 4.85 were reported for (*E*)-*N*′-((2-hydroxy-6-methylquinolin-3-yl)methylene)-4 methylbenzenesulfonohydrazide complexes with Cu(II), Ni(II) and Co(II) [14].

### 2.6. FTIR Spectra of Metal Complexes

The FTIR spectrum of free NQ ligand shows sharp strong bands at 1645, 1619 and 1599 cm$^{-1}$ (Table 3) which are assigned to the stretching vibration of carbonyl $\nu$(C=O), azomethine $\nu$(-HC=N-) and quinoline $\nu$(C=N), respectively, according to [24]. Although there is a variance in the reported azomethine C=N stretching frequency depending on the structures of Schiff bases which was reported at 1636 [15], 1620 [18] and 1602 cm$^{-1}$ [19], the bond lengths obtained from crystallographic data (Table S6, 1.232, 1.277 and 1.323 Å, respectively) supported our assignment. The three absorption bands were shifted to a higher frequency upon complexation with Cu(II), Cd(II), Cr(III) and Fe(III), among which the most pronounced shift was in the case of Fe(III). A reverse shift to a lower frequency was observed in the case of Co(II) and Ni(II). Moreover, two weak bands in the range 466–477 cm$^{-1}$ and 500–565 cm$^{-1}$ were assigned to metal–nitrogen and metal–oxygen stretching vibrations, respectively (Table 3). Patil and Vibhute reported close regions 463–468 cm$^{-1}$ and 506–516 cm$^{-1}$, respectively, for Schiff base complexes with Cu, Ni, Co and Cd [14]. Furthermore, Althobiti and Zabin reported similar regions for Cu-N and Cu-O stretching vibrations, namely, 451–488 cm$^{-1}$ and 542–586 cm$^{-1}$ [16].

**Table 3.** IR spectral bands (cm$^{-1}$) of the Schiff base ligand (NQ) and its metal complexes.

| Code | $\nu$(C=O) | $\nu$(-HC=N-)<br>Azomethine | $\nu$(-C=N-)<br>Quinoline | M-O | M-N |
|---|---|---|---|---|---|
| NQ | 1645 | 1619 | 1599 | | |
| Cu-NQ | 1693 | 1638 | 1620 | 564 | 475 |
| Cd-NQ | 1668 | 1639 | 1621 | 565 | 477 |
| Cr-NQ | 1686 | 1639 | 1601 | 500 | 474 |
| Fe-NQ | 1705 | 1683 | 1669 | 518 | 471 |
| Co-NQ | 1622 | 1603 | 1565 | 545 | 466 |
| Ni-NQ | 1623 | 1594 | 1564 | 538 | 468 |

Some of the characteristic FTIR spectral bands of the CQ ligand and its complexes are shown in Figure 3 and Table 4. Similar to the case of NQ, the bands at 1641, 1614 and 1595 cm$^{-1}$ in the spectrum of CQ were assigned to $\nu$(C=O), azomethine $\nu$(-HC=N-) and quinoline $\nu$(C=N), respectively. The carbonyl and azomethine bands were shifted to higher frequencies in the spectra of metal complexes. On the other hand, the quinoline C=N was not affected by complexation. Thus, the FTIR data support coordination of ligand via carbonyl oxygen and azomethine nitrogen. However, the shift of azomethine stretching vibration of NQ and CQ ligands to a higher frequency upon complexation with metal ions is strange since most previous works reported lower frequency shifts [13–15,18,19]. The reason for the higher frequency shift observed in the present work may be the coordination of metal ions to the oxygen of the carbonyl group which facilitates predominance of resonance form I over III. Since III weakened the azomethine C=N bond, the predominance of form I will cause shifting to a higher frequency.

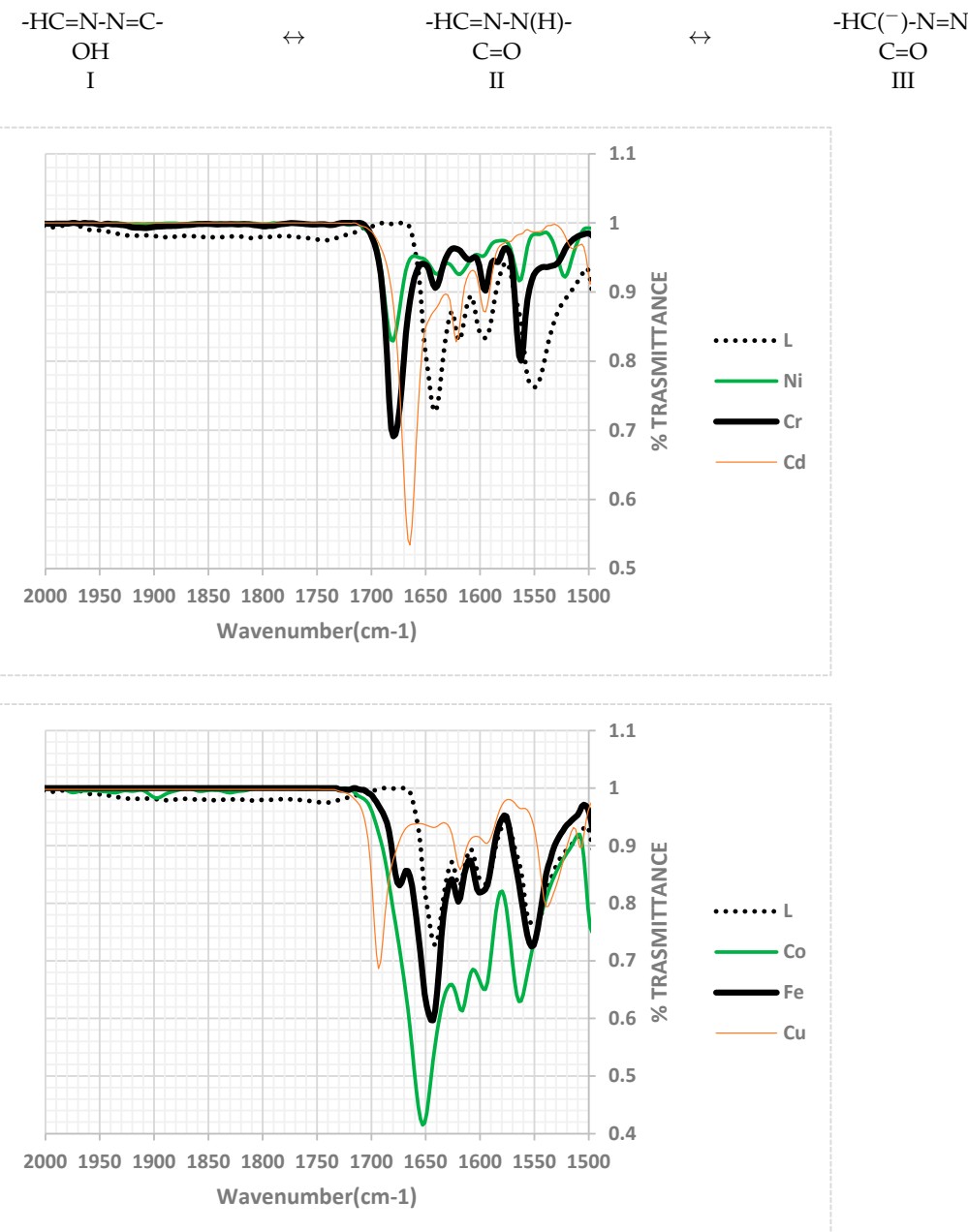

**Figure 3.** FTIR spectra of CQ complexes with metal ions in the range from 1500 to 2000 cm$^{-1}$.

**Table 4.** FTIR main stretching absorption bands (cm$^{-1}$) of CQ and its metal complexes.

| Code | ν(C=O) | ν(-HC=N-) Azomethine | ν(-C=N-) Quinoline | M-O | M-N |
|---|---|---|---|---|---|
| CQ | 1641 | 1619 | 1595 | | |
| Cu-CQ | 1693 | 1643, 1618 | 1594 | 528 | 436 |
| Cd-CQ | 1665 | 1639, 1622 | 1596 | 529 | 504 |
| Cr-CQ | 1679 | 1640 | 1595 | 526 | 516 |
| Fe-CQ | 1644 | 1620 | 1600 | 535 | 454 |
| Co-CQ | 1651 | 1617 | 1596 | 529 | 432 |
| Ni-CQ | 1682 | 1643, 1618 | 1597 | 503 | 441 |

The FTIR spectral bands of the HQ ligand and its complexes are shown in Figure 4 and Table 5. In contrast to NQ and CQ ligands discussed above, the FTIR absorption bands at 1660, 1630 and 1608 cm$^{-1}$ in the spectrum of CQ, which were assigned to the stretching vibration of carbonyl (C=O), azomethine ν(-HC=N-) and quinoline ν(C=N), respectively, were shifted to a lower frequency upon complexation with metal ions. The shift of the azomethine C=N to a lower frequency is consistent with those reported for Schiff bases containing an azomethine group attached to an aromatic ring in the ortho position to a phenolic group [14,15,18]. Thus, the shift in the case of HQ is in the reverse direction of the shift in the case of NQ and CQ, suggesting the HQ is coordinated to the metal ion via azomethine nitrogen and phenolic oxygen rather than through azomethine nitrogen and carbonyl oxygen as in the case of NQ and CQ.

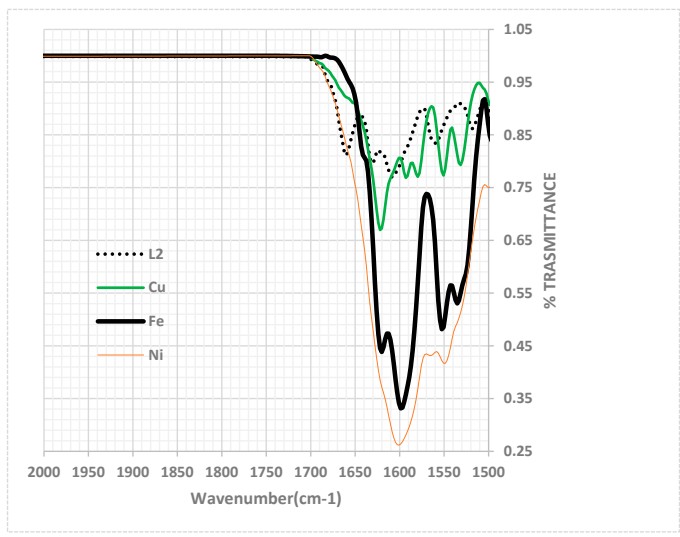

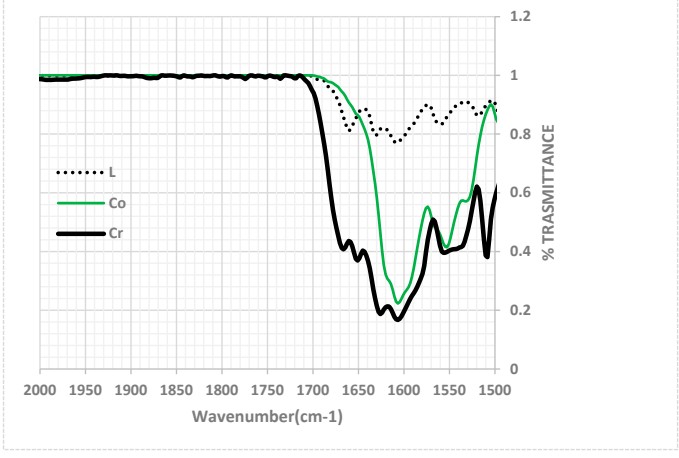

**Figure 4.** FTIR of HQ with metal complexes in the range from 1500 to 2000 cm$^{-1}$.

**Table 5.** FTIR main stretching absorption bands (cm$^{-1}$) of HQ and its metal complexes.

| Code | ν(C=O) | ν(-HC=N-) Azomethine | ν(-C=N-) Quinoline | ν(C-OH) Phenolic | M-O | M-N |
|---|---|---|---|---|---|---|
| HQ | 1660 | 1630 | 1608 | 1493, 1455 | | |
| Cu-HQ | 1621 | 1592 | 1579 | 1497, 1443 | 600 | 505 |
| Cd-HQ | 1646 | 1633 | 1612 | 1457, 1404 | 587 | 503 |
| Cr-HQ | 1650 | 1624 | 1605 | 1456, 1438 | 593 | 533 |
| Fe-HQ | 1620 | 1598 | 1551 | 1455, 1436 | 592 | 511 |
| Co-HQ | 1606 | 1618 | 1528 | 1483, 1444 | 548 | 508 |
| Ni-HQ | 1601 | 1565 | 1549 | 1440, 1401 | 594 | 507 |

*2.7. $^1$H-NMR Study of Complexes*

Due the paramagnetic nature of the complexes (Table 6), the complexes exhibited broadened and noisy $^1$H-NMR spectra (Figures S3–S12), except Cd-CQ (diamagnetic) (Figure S12) and Co-HQ (Figure S6) complexes. The Co-HQ complex exhibited sharp peaks similar to those of the free ligand, which reflects a low paramagnetic field as evidenced also by the low magnetic susceptibility of this complex (Table 6). However, it was possible to follow the effect of metal complexation on the N-H, azomethine HC=N and quinoline HC=N peaks (Figures S3–S12) in some complexes as illustrated in Table 6. It can be concluded that the azomethine HC=N proton at 9.3 ppm is the most affected by complexation, indicating that the azomethine nitrogen site in the ligands is coordinated to the metal ion in the solution, despite the high solvation power of DMSO solvent. Quinoline nitrogen was reported to be an effective coordinating site for metal ions when the azomethine group is in the 1-position of quinoline [13,19–21,36]. In the present work, the quinoline nitrogen is either far from the azomethine nitrogen or strongly solvated by DMSO solvent.

**Table 6.** Some $^1$H-NMR peaks of NQ, CQ and HQ and their metal complexes (ppm, assignment of peaks according to ref. [24]).

| | -N-H | -HC=N- Azomethine | Quinoline (H1,3) | O-H |
|---|---|---|---|---|
| NQ | 12.54 | 9.36 | 8.98, 8.93 | |
| Cu-NQ | 12.56 | 9.84 (broad) | 9.02 (broad) | |
| CQ | 12.43 | 9.36 | 8.98, 8.92 | |
| Cd-CQ | 12.42 | 9.37 | 8.98, 8.92 | |
| Ni-CQ | 12.47 | 9.43 (broad) | 8.98, 8.92 | |
| HQ | 12.23 | 9.34 | 8.92, 8.57 | 11.36, 10.01 |
| Co-HQ | 12.22 | 9.72 (broad), 9.36 | 8.91, 8.51 | 11.35, 10.00 |
| Ni-CQ | 12.33 | 12.69 (broad) | 9.00, 8.59 | 11.36 |

*2.8. Biological Activity of Ligands and Complexes (Antiproliferative Activity)*

For the human breast adenocarcinoma MCF-7 line, among the compounds tested, Co-CQ, CQ, Cu-CQ and Fe-CQ showed limited antiproliferative activity, with IC50 values greater than 50.00 μM (Table 7, Figure S13). This indicates that these compounds are not effective in inhibiting cell growth within the specified concentration range. On the other hand, the following compounds exhibited notable inhibitory effects on the MCF7 line, in descending potency order: Cu-HQ (1.19 ± 0.03 μM), Ni-HQ (5.85 ± 0.15 μM), HQ (7.19 ± 0.18 μM), Cu-NQ (17.47 ± 0.44 μM), Ni-NQ (26.36 ± 0.66 μM), Co-HQ (27.65 ± 0.69 μM) and NQ (37.8 ± 0.94 μM) (Table 7, Figure S13).

**Table 7.** The 50% inhibitory concentration (IC50) values for tested compound treatment in MCF7, K562 and fibroblast cell lines. Experiments were carried out for a 72 h treatment duration. Experiments were run in triplicates for at least three independent trials (*n* = 6). Concentrations are expressed in micromoles. NA: not applicable; SD: standard deviation; h: hour; µM: micromoles.

| Cell Line | MCF7 | K562 | Fibroblast |
|---|---|---|---|
| NQ | $37.8 \pm 0.94$ | $41.57 \pm 1.04$ | >50.00 |
| Cu-NQ | $17.47 \pm 0.44$ | $17.13 \pm 0.43$ | $42.42 \pm 1.06$ |
| Ni-NQ | $26.36 \pm 0.66$ | $57.42 \pm 1.44$ | >50.00 |
| CQ | >50.00 | >50.00 | >50.00 |
| Cu-CQ | >50.00 | >50.00 | >50.00 |
| Fe-CQ | >50.00 | >50.00 | $28.5 \pm 0.71$ |
| Co-CQ | >50.00 | $29.65 \pm 0.74$ | >50.00 |
| HQ | $7.19 \pm 0.18$ | $2.03 \pm 0.05$ | >50.00 |
| Cu-HQ | $1.19 \pm 0.03$ | $8.18 \pm 0.2$ | $77.6 \pm 1.94$ |
| Fe-HQ | >50.00 | >50.00 | >50.00 |
| Co-HQ | $27.65 \pm 0.69$ | $18.12 \pm 0.45$ | >50.00 |
| Ni-HQ | $5.85 \pm 0.15$ | $2.32 \pm 0.06$ | >50.00 |

For the human chronic myelogenous leukemia K562 line, among the compounds tested, CQ, Cu-CQ, Fe-CQ and Fe-HQ showed limited antiproliferative activity, with IC50 values greater than 50.00 µM. On the other hand, HQ, Ni-HQ, Cu-HQ, Cu-NQ, Co-HQ, NQ, Ni-NQ exhibited notable inhibitory effects on the K562. The descending potency order was: HQ ($2.03 \pm 0.05$ µM), Ni-HQ ($2.32 \pm 0.06$ µM), Cu-HQ ($8.18 \pm 0.2$ µM), Cu-NQ ($17.13 \pm 0.43$ µM), Co-HQ ($18.12 \pm 0.45$ µM) and NQ ($41.57 \pm 1.04$ µM) (Table 7, Figure S14).

In the case of a human dermal fibroblast cell line, most compounds showed limited antiproliferative activity, with IC50 values greater than 50.00 µM, except Fe-CQ ($28.5 \pm 0.71$ µM) and Cu-NQ ($42.42 \pm 1.06$ µM) (Table 7).

From results in Table 7, it is clear that the presence of two phenolic groups (electron-donating group) in HQ induces the activity more than electron withdrawing groups/atoms, namely nitro (NQ) or chloro (CQ). Relatively, the nitro group is better than chlorine atoms regarding the activity. The phenolic and, to a lesser extent, nitro group exhibit hydrogen bonding interaction as hydrogen bond acceptors. Phenolic compounds are known for their anti-inflammatory, antimicrobial, antioxidant and anticarcinogenic activity [37].

## 3. Materials and Methods

### 3.1. Materials and Instruments

The organic precursor ethyl-3-quinoline carboxylate was purchased from Sigma-Aldrich (Darmstadt, Germany). 2-Chlorebenzaldehyde, 2-nitrobenzaldehyde and 2,4-dihydroxybenzaldehyde were from Fluka (Darmstadt, Germany) and Acros Organics (Waltham, Massachusetts, United States), respectively. $CuCl_2 \cdot 2H_2O$, $NiCl_2 \cdot 6H_2O$ and $CoCl_2 \cdot 6H_2O$ were obtained from Sigma-Aldrich. $CdCl_2 \cdot H_2O$ and $CrCl_3 \cdot 6H_2O$ were from Acros Organics. $FeCl_3 \cdot 6H_2O$ was from Riedel-de Haën (Hanover, Germany). The melting points of synthesized compounds were determined using Stuart melting apparatus (Stuart, UK). Elemental analysis was carried out using an automated elemental analyzer (EuroEA3000, Pavia, Italy). The magnetic susceptibility measurements were conducted at room temperature ($25 \pm 2$ °C) using a Johnson–Matthey balance (Johnson-Matthey Com-

pany, Germany). The effective magnetic moment (μeff) was calculated using Equation (1), where Xm is the molar magnetic susceptibility ($m^3$/mol) and T is the temperature (°C).

$$\mu eff = 2.84\sqrt{(Xm\ T)} \tag{1}$$

Fourier transform infrared (FTIR) spectra were measured in the region between 4000 and 400 $cm^{-1}$ using the KBr disc method on a Bruker Vertex 70-FTIR spectrometer (Bruker, Fällanden, Switzerland). NMR spectra were recorded using an AVANCE-III 400 MHz FT-NMR NanoBay spectrometer (Bruker, Fällanden, Switzerland) in dimethyl sulfoxide (DMSO-$d_6$) with tetramethyl silane (TMS) as an internal standard.

X-ray crystallographic data were collected on an Oxford Diffraction Gemini CCD diffractometer with Cu-Kα radiation for CQ·$H_2O$ (1.54184 Å) or Mo-Kα radiation for NQ·$H_2O$ (0.71073 Å). The samples were cooled to 190 K with an Oxford Cryosystems Desktop Cooler. Data reduction and empirical absorption corrections were performed using CrysAlisPro (version 171.42.49, Rigaku Oxford Diffraction). The structures were solved by Direct Methods by SHELXT and refined with SHELXL [38]. All calculations were carried out within the WinGX package [39]. All non-H-atoms were refined anisotropically. The data in CIF format have been deposited with the CCDC (numbers 2279657 and 2279658).

### 3.2. General Procedure for Synthesis Quinoline-3-Carbohydrazide (PQ)

The precursor quinoline-3-carbohydrazide (PQ) was synthesized as illustrated (Scheme 1) through a modified procedure [24]. Ethyl-3-quinoline carboxylate was converted to quinoline-3-carbohydrazide via refluxing 4.969 mmol of ethyl-3-quinoline carboxylate with 19.877 mmol of hydrazine hydrate (1000 mg) for 24 h. The reaction mixture was then left overnight at room temperature to allow it to cool down and to precipitate. The product was filtered, washed with ethanol and dried, then recrystallized with hot ethanol to yield (92%) a white powder, m.p. (184–186 °C).

### 3.3. Synthesis of Schiff Base Ligand

Synthesis of Schiff base ligands (NQ, CQ and HQ) is illustrated in Scheme 1. Condensations of precursor 3-quinoline carbohydrazide (PQ) with 2-nitrobenzaldehyde, 2-chlorebenzaldehyde and 2,4-dihydroxybenzaldehyed, respectively, were carried out using the modified method of [24]. The aldehyde (5.548 mmol) was dissolved in 30 mL of ethanol and 3 drops of acetic acid were added and then mixed with an ethanolic solution of PQ (5.548 mmol). After refluxing the mixture for 5 h, the solvent was evaporated until a precipitate was obtained. The precipitate was collected via suction filtration and recrystallized using hot ethanol and dried at 60 °C in a vacuum oven.

#### 3.3.1. Synthesis of (E)-N'-(2-Nitrobenzylidene) Quinoline-3-Carbohydrazide, NQ

An 85% yield of a pale-yellow powder, m.p. = 235–237 °C, elemental analysis for [$C_{17}H_{12}N_4O_3$]: found (calculated) %: C 63.71 (63.75), H 3.58(3.78), N 17.32 (17.49). FTIR data (KBr disc, $cm^{-1}$): 3196v (-NH- amine), 1645v (C=O), 1619v (azomethine -HC=N-), 1599v (-C=N- quinoline), 1522v (amide -OCNH-), 1341v ($NO_2$). $^1$H NMR (400 MHz, DMSO-$d_6$) δ 12.54 (s, 1H, NH), 9.36 (s, 1H, azomethine -HC=N-), 8.98 (s, 1H), 8.93 (s, 1H, quinoline H1 and H3), 8.22–8.09 (m, 4H), 7.97–7.83 (m, 2H), 7.78–7.70 (m, 2H). $^{13}$C-NMR (100 MHz, DMSO-$d_6$) δ 162.46 (C=O), 149.42 (-C=N- quinoline), 149.18, 148.77, 144.13 (azomethine -HC=N-), 136.78, 134.32, 132.10, 131.36, 129.71, 129.32, 129.10, 128.54, 128.11, 126.88, 126.29, 125.22, which were ascribed to the aromatic carbons; 2D-NMR which includes COSY, HMQC and HMBC was used for the assignment of $^1$H and $^{13}$C-NMR chemical shift of NQ ligand.

#### 3.3.2. Synthesis of (E)-N'-(2-Chlorobenzylidene) Quinoline-3-Carbohydrazide, CQ

An 86% yield of a pale-yellow powder, m.p. = 223–225 °C, elemental analysis for [$C_{17}H_{12}ClN_3O$]: found (calculated) %: C 65.48 (65.92), H 3.89 (3.90), N 13.66 (13.57). FTIR

data (KBr disc, cm$^{-1}$): 3176v (-NH- amide), 1641v (C=O), 1619v (azomethine -HC=N-), 1595v (-C=N- quinoline), 741v (C-Cl). $^1$H NMR (400 MHz, DMSO-d$_6$) δ/ppm: 12.43 (s, 1H, -NH-), 9.36 (s, 1H,) azomethine -HC=N-, 8.98 (s, 1H), 8.92 (s, 1H, quinoline H1 and H3), 8.21–8.04 (m, 3H), 7.93 (t, J$^3$ = 7.5 Hz, 1H), 7.75 (t, J$^3$ = 7.4 Hz, 1H), 7.61–7.42 (m, 3H). $^{13}$C-NMR (100 MHz, DMSO-d$_6$) δ: 162.27(C=O), 149.39, 149.33(-C=N- quinoline), 144.71 (-HC=N- azomethine), 136.70, 133.80, 132.21, 132.07, 131.88, 130.47, 129.69, 129.32, 128.20, 128.11, 127.46, 126.91, 126.44, which were assigned to the aromatic carbons.

### 3.3.3. Synthesis of (E)-N'-(2,4-Dihydroxybenzylidene) Quinoline-3-carbohydrazide, HQ

An 80% yield of a pale-yellow powder, m.p. = 288-290 °C (dec), elemental analysis for [C$_{17}$H$_{13}$N$_3$O$_3$]: found (calculated) %: C 66.39 (66.44); H 4.29 (4.26); N 13.69 (13.67). FTIR data (KBr disc, cm$^{-1}$): 3252v (-NH- amide), 1660v (C=O), 1630v (azomethane -HC=N-), 1608v (-C=N- quinoline), 1248v (-C-OH phenolic). $^1$H NMR (400 MHz, DMSO-d$_6$) δ/ppm: 12.23 (s, 1H, -NH), 11.36 (s, 1H, OH), 10.01 (s, 1H, OH), 9.34 (s, 1H, azomethine -HC=N-), 8.92 (s, 1H, quinoline H1 and H3), 8.57 (s, 1H), 8.27–7.99 (m, 2H), 7.91 (t, J$^3$ = 7.6 Hz, 1H), 7.73 (t, J$^3$ = 7.5 Hz, 1H), 7.38 (d, J$^3$ = 8.4 Hz, 1H), 6.48–6.29 (m, 2H). $^{13}$C-NMR (100 MHz, DMSO-d$_6$) δ: 161.56 (C=O), 161.41(C-OH), 159.99 (C-OH), 149.86, 149.32(-C=N- quinoline), 149.09 (azomethane -HC=N-), 136.49, 131.96, 131.69, 129.65, 129.30, 128.06, 126.93, 126.38, 110.99, 108.29, 103.14, which were assigned to the aromatic carbons.

A silica gel sheet (20 cm × 20 cm, Merck Aluminum, Germany) and 6:4 hexane–ethyl acetate mobile phase were used for thin layer chromatographic monitoring of synthesis of ligands and visualization of the sheet was conducted under a UV lamp (354 nm).

### *3.4. Synthesis of Metal Complexes*
### 3.4.1. Synthesis of NQ Complexes

The NQ complexes were prepared by refluxing ethanolic solutions of 0.66 mmol of the NQ ligand with 0.33 mmol of the metal chloride (CuCl$_2$·2H$_2$O, CdCl$_2$·H$_2$O, CrCl$_3$·6H$_2$O, FeCl$_3$·6H$_2$O, NiCl$_2$·6H$_2$O, CoCl$_2$·6H$_2$O) for 1 h. The mixture was evaporated until a precipitate was formed, which was collected by suction filtration, washed with ethanol and dried in a vacuum oven at 60 °C. The physical properties for NQ complexes: yield, color, melting point, and elemental analysis data are:

- Cu-NQ: 67% yield, green color, m.p.: 220–222 °C, elemental analysis %: C 52.35, H 3.48, N 14.46, calculated % for CuL$_2$Cl(OH)·H$_2$O: C 52.72, H 3.51, N 14.47.
- Cd-NQ: 72% yield, reddish-brown color, m.p.: >300 °C, elemental analysis %: C 54.35, H 3.52, N 14.92, calculated % for CdL$_3$(OH)$_2$·H$_2$O: C 54.43, H 3.58, N 14.94.
- Cr-NQ: 68% yield, green color, m.p.: 283–285 °C (dec), elemental analysis %: C 51.66, H, 2.99, N 14.22, calculated % for CrL$_2$Cl$_2$(OH)·H$_2$O: C 51.14, H 3.41, N 14.03.
- Fe-NQ: 66% yield, black color, m.p.: 78–80 °C, elemental analysis %: C 50.90, H, 3.47, N 13.86, calculated % for FeL$_2$Cl$_2$(OH)·H$_2$O: C 50.89, H 3.39, N 13.97.
- Co-NQ: 67% yield, blue color, m.p.: 100–102 °C, elemental analysis %: C 53.57, H 3.78, N 14.95, calculated % for CoL$_2$Cl(OH)·H$_2$O: C 53.03, H 3.53, N 14.55.
- Ni-NQ: 59% yield, reddish-brown color, m.p.: >300 °C, elemental analysis %: C 53.05, H 3.09, N 14.49, calculated % for NiL$_2$Cl$_2$**:** C 53.02, H 3.14, N 14.55.

### 3.4.2. Synthesis of CQ Complexes

The CQ complexes were prepared according to the procedure described for NQ complexes. The physical properties for CQ complexes: yield, color, melting point, and elemental analysis data are:

- Cu-CQ: 58% yield, green color, m.p.: 233–235 °C, elemental analysis %: C, 54.29, H 3.57, N 11.94, calculated % for CuL$_2$Cl(OH)·H$_2$O: C 54.19, H 3.61, N 11.15.
- Cd-CQ: 70% yield, yellow color, m.p.: >300 °C, elemental analysis %: C 55.94, H 3.77, N 11.62, calculated % for CdL$_3$(OH)$_2$·H$_2$O: C 56.01, H 3.69, N 11.53.
- Cr-CQ: 65% yield, green color, m.p.: 235–237 °C, elemental analysis %: C 52.72, H 3.26, N 11.02, calculated % for CrL$_2$Cl$_2$(OH)·H$_2$O: C 52.53, H 3.50, N 10.81.

- Fe-CQ: 69% yield, yellow-green color, m.p.: 204–206 °C, elemental analysis %: C 52.45, H 3.25, N 10.65, calculated % for $FeL_2Cl_2(OH) \cdot H_2O$: C 52.27, H 3.48, N 10.76.
- Co-CQ: 62% yield, green color, m.p.: 220–222 °C, elemental analysis %: C 54.89, H 3.81, N 11.97, calculated % for $CoL_2Cl(OH) \cdot H_2O$: C 54.53, H 3.63, N 11.22.
- Ni-CQ: 66% yield, orange color, m.p.: 240–242 °C, elemental analysis %: C 54.25, H 3.01, N 10.91, calculated % for $NiL_2Cl_2$: C 54.51, H 3.23, N 11.22.

### 3.4.3. Synthesis of HQ Complexes

The HQ complexes were prepared according to the procedure described for NQ and CQ complexes. The physical properties for HQ complexes and elemental analysis data are:

- Cu-HQ: 62% yield, dark-yellow color, m.p.: 222–224 °C, elemental analysis % C 53.92, H 3.20, N, 10.75, calculated % for $CuL_2Cl(OH) \cdot H_2O$: C 54.55, H 3.90, N 11.23.
- Cd-HQ: 72% yield, pale-yellow color, m.p.: 273–275 °C, elemental analysis %: C 56.40, H 3.78, N, 11.75, calculated % for $CdL_3(OH)_2 \cdot H_2O$: C 56.39, H 3.99, N 11.60.
- Cr-HQ: 69% yield, orange color, m.p.: 108–110 °C, elemental analysis %: C, 56.73, H 3.59, N 11.47, calculated % for $CrL_2Cl(OH)_2$: C, 55.48, H, 3.83; N, 11.42.
- Fe-HQ: 65% yield, brown color, m.p.: >300 °C, elemental analysis: C 52.80, H 3.70, N 10.75, calculated % for $FeL_2Cl_2(OH) \cdot H_2O$: C 52.60, H 3.76, N 10.82.
- Co-HQ: 69% yield, reddish-brown color, m.p.: >300 °C, elemental analysis %: C 54.90, H 3.65, N 11.95, calculated % for $CoL_2Cl(OH) \cdot H_2O$: C 54.89, H 3.93, N 11.30.
- Ni-HQ: 66% yield, orange color, m.p.: 285–287 °C, elemental analysis %: C 56.10, H 3.61, N 10.5, calculated % for $NiL_2Cl(OH)$: C 56.27, H 3.75, Cl 4.88, N 11.58.

### 3.5. Antiproliferative Activity

#### 3.5.1. Cells and Cell Culture Conditions

The human breast adenocarcinoma MCF-7, the human chronic myelogenous leukemia K562 and the human dermal fibroblast cell lines were purchased from the American Type Culture Collection (Rockville, MD, USA). Cells were grown and maintained with appropriate media, RPMI medium (EuroClone, Italy) was used to grow MCF-7 and K562, while Iscove's modified Dulbecco's medium (IMDM, EuroClone, Italy) was used to grow fibroblasts. All media were supplemented with 10% fetal bovine serum (FBS), 2mM L-glutamine and penicillin (100 U/mL) and were incubated at 37 °C in a humidified atmosphere of 95% $O_2$ and 5% $CO_2$. Cell cultures were passaged every 2–3 days or whenever reaching 80% confluency. Cells were seeded at a density of $6$–$8 \times 10^3$ cells/well in 96-well plates and incubated for 24 h for adhesion.

#### 3.5.2. Cell Proliferation Assay (MTT)

The MTT colorimetric assay was employed to evaluate cell proliferation as previously described [40]. In brief, test samples were prepared by dissolution into molecular grade dimethyl sulfoxide (DMSO) followed by further dilution with relevant media to reach the desired final concentration. The final DMSO concentration in the assay was kept as low as 0.1%. Screening for potential antiproliferative effects was carried out using two concentration points in triplicates, 10 μM and 50 μM, for 72 h. Nevertheless, all compounds were further evaluated for their antiproliferative effect on a wider range of concentration across all study cell lines. All experiments were run in triplicates and for at least three independent trials. After exposing cell lines to treatment for 72 h, MTT solution was applied to each well and incubated for 4 h at 37 °C. Afterwards, DMSO was added to each well to solubilize the purple formazan crystals formed. Then, absorbance was read using a microplate reader at 570 nm.

#### 3.5.3. Statistical Analysis

All experiments were conducted in triplicates and results are expressed as mean ± standard deviation (SD). IC50 for antiproliferative activity was calculated us-

ing nonlinear regression by GraphPad Prism software (GraphPad Prism version 9.0.0 for Windows, GraphPad Software, San Diego, CA, USA).

## 4. Conclusions

Three Schiff base ligands, NQ, CQ and HQ, were prepared from the reaction of quinoline-3-carbohydrazide (PQ) with 2-nitrobenzaldehyde, 2-chlorobenzaldehyde and 2,4-dihydroxybenzaldehyde, respectively. The NQ ligand preparation and the X-ray structure of NQ and CQ, as well as transition metal complexes of NQ, CQ and HQ, were reported for the first time. Depending on the magnetic susceptibility and elemental analysis results, octahedral structures of the complexes such as $[CuL_2Cl(OH)]$, $[FeL_2Cl_2(OH)]$ and $[CoL_2Cl(OH)]$ were proposed for L = NQ, CQ and HQ. The relatively large Cd(II) exhibited $[CdL_3(OH)_2]$.

As evidenced by FTIR spectroscopy, in the solid state, HQ is coordinated to the metal ion via azomethine nitrogen and phenolic oxygen. On the other hand, NQ and CQ are coordinated through azomethine nitrogen and carbonyl oxygen. $^1$H-NMR experiments indicated that the azomethine site in the ligands is the most preferable site for coordination to metal ion in the solution. Furthermore, HQ ligand and its complexes with Cu and Ni exhibited notable inhibitory effects on human breast adenocarcinoma MCF-7 and human chronic myelogenous leukemia K562. This distinguished reactivity for HQ ligand and its complexes may be due to the phenolic groups of HQ. In the complexes, metal ions did not exhibit a significant synergistic effect on the antiproliferative activity of the ligands investigated.

**Supplementary Materials:** The following supporting information can be downloaded at: https://www.mdpi.com/article/10.3390/inorganics11100412/s1.

**Author Contributions:** Conceptualization, M.S. (Mahmoud Sunjuk) and M.S. (Majed Shtaiwi); methodology, L.A.-N. and M.S. (Majed Shtaiwi); software, K.S. and P.V.B.; validation, M.S. (Mahmoud Sunjuk), M.S. (Majed Shtaiwi) and L.A.-N.; formal analysis, P.V.B. and H.Z.; investigation, M.S. (Majed Shtaiwi) and H.Z.; resources, L.A.-E.; data curation, L.A.-N. and K.S.; writing—original draft preparation, B.I.E.-E. and L.A.-N.; writing—review and editing, L.A.-E. and B.I.E.-E.; visualization, P.V.B. and K.S.; supervision, M.S. (Mahmoud Sunjuk) and M.S. (Majed Shtaiwi); project administration, M.S. (Mahmoud Sunjuk); funding acquisition, M.S. (Mahmoud Sunjuk). All authors have read and agreed to the published version of the manuscript.

**Funding:** This research received no external funding.

**Data Availability Statement:** Data is contained within the article or supplementary material.

**Conflicts of Interest:** The authors declare no conflict of interest.

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
