# Peer review of "Metal Complexes of Schiff Bases Prepared from Quinoline-3-Carbohydrazide with 2-Nitrobenzaldehyde, 2-Chlorobenzaldehyde and 2,4-Dihydroxybenzaldehyde: Structure and Biological Activity"

_inorganics, doi:10.3390/inorganics11100412_

Round 1

Reviewer 1 Report

The manuscript is well documented and drafted. The study has some limitations, since a small number of compounds are presented and the degree of originality is low, as the complexation ability of Schiff bases is well known. However, the experimental data are well presented. The synthesized ligands and complexes are properly characterized, both concerning the structure characterization as well as the magnetic susceptibility (for metal complexes). The importance of the magnetic susceptibility studies is no doubt, of high impact, since the presented data are compared with some previously obtained results. In conclusion, after minor modifications according to the recommendation below, the article could be accepted for publication in Inorganics.

1. Replace “…method of synthesis of Schiff bases.” with “…method for synthesis of Schiff bases” (page 2, line 1).

2. Since, neither the Schiff bases nor their corresponding metal complexes are inorganic compounds, please rephrase: Schiff bases are very common ligands in inorganic chemistry…” (page 2, line 2) (for example: Schiff bases are very common ligands in organometallic chemistry).

3. Replace “…antifungal activities[6], [7].” with “…antifungal activities [6], [7].” (page 2, line 7).

4. Add space after the parenthesis: “…(IC50 of 4 and 3.2 µM, respectively)[13].” (page 2, line 22)

5. Replace “Patil et al” with “Pati et al.” (page 2, line 22), like previously format used (“Adsule et al.”, page 2, line 20)

6. Use Italic style for both, stereoisomers (E) and heteroatoms (N, O,…) in all the manuscript (for example, replace: “(E)-N’-((2-hydroxyquinolin-3-yl) methylene)-4-methylbenzenesulfonohydrazide)” with “(E)-N’-((2-hydroxyquinolin-3-yl)methylene)-4-methylbenzenesulfonohydrazide)”, page 2, line 23).

7. Use Italic style for “in vitro” in all the manuscript (for example, page 2, line 25,….)

8. Replace “ml” with “mL” (page 2, Scheme 1)

9. Avoid double repetition of stretching vibration value azomethine group ν(-HC=N-)  (1630 cm-1, page 3, line 14). The value is clearly observed in Table 1 for both, CQ and HQ compounds.

10. Avoid double repetition of “…in in…” (page 3, line 17)

11. Replace “DMSO-d6” with “DMSO-d6” in all the manuscript (for example, page 3, line 1, below Table 1)

12. Please correct in all the manuscript the symbol for Celsius degree by using Symbol Insert function (°C) (for example, page 5, line 2, above Scheme 2)

13. Replace “via” with “via”(page 12, paragraph 3.3)

14. Replace “Merk” with “Merck” (page 13, paragraph 3.3.3.)

15. How can you explain the satisfactory but not very high yields of the NQ/CQ/HQ complexes?

16. Since the manuscript is entitled: “Metal complexes of Schiff bases…. structure and biological activity”, more details about the antiproliferative activity of the tested compounds should by presented in paragraph 3.5., even the figures 13S and 14S (from supplementary material) could be inserted here.

The English language is satisfactory.

Author Response

REVIEWER #1

The manuscript is well documented and drafted. The study has some limitations, since a small number of compounds are presented and the degree of originality is low, as the complexation ability of Schiff bases is well known.

Response: The originality points were added to the abstract, line 23: “The NQ preparation and the X-ray structure of NQ and CQ, as well as the transition metal complexes of NQ, CQ and HQ were reported for the first time”

Furthermore, the degree of originality was clarified by comparing the Schiff base complexes of the present work with those in the literature

lines 68: “In the literature, the azomethine nitrogen is conjugated with phenolic oxygen [14]–[18], quinoline [19]–[22] or phenolic oxygen and carbonyl [23], [24] to facilitate the chelating/coordinating properties of ligands to metal ions. The present work will describe the synthesis of Cu(II), Co(II), Ni(II), Cd(II), Cr(III) and Fe(III) complexes with three Schiff base ligands (NQ, CQ and HQ) which were prepared by condensation of Quinoline-3-carbohydrazide with aromatic aldehyde: 2-nitrobenzaldehyde, 2-chlorebenzaldehyde and 2,4-dihydroxybenzaldehyed, respectively (Scheme1). The ligands have an array of quinolone nitrogen, carbonyl oxygen, and azomethine nitrogen, as well as phenolic oxygen (in the case of HQ), rendering potential coordination sites to metal ions.”

However, the experimental data are well presented. The synthesized ligands and complexes are properly characterized, both concerning the structure characterization as well as the magnetic susceptibility (for metal complexes). The importance of the magnetic susceptibility studies is no doubt, of high impact, since the presented data are compared with some previously obtained results.

In conclusion, after minor modifications according to the recommendation below, the article could be accepted for publication in Inorganics.

  1. Replace “…method of synthesis of Schiff bases.” with “…method for synthesis of Schiff bases” (page 2, line 1).

Response: Corrected.

  1. Since, neither the Schiff bases nor their corresponding metal complexes are inorganic compounds, please rephrase: “Schiff bases are very common ligands in inorganic chemistry…” (page 2, line 2) (for example: Schiff bases are very common ligands in organometallic chemistry).

Response: Corrected.

  1. Replace “…antifungal activities[6], [7].” with “…antifungal activities [6], [7].” (page 2, line 7).

Response: Corrected.

  1. Add space after the parenthesis: “…(IC50 of 4 and 3.2 µM, respectively)[13].” (page 2, line 22)

Response: Corrected.

  1. Replace “Patil et al” with “Pati et al.” (page 2, line 22), like previously format used (“Adsule et al.”, page 2, line 20)

Response: Corrected.

  1. Use Italic style for both, stereoisomers (E) and heteroatoms (N, O,…) in all the manuscript (for example, replace: “(E)-N’-((2-hydroxyquinolin-3-yl) methylene)-4-methylbenzenesulfonohydrazide)” with “(E)-N’-((2-hydroxyquinolin-3-yl)methylene)-4-methylbenzenesulfonohydrazide)”, page 2, line 23).

Response: Corrected.

  1. Use Italic style for “in vitro” in all the manuscript (for example, page 2, line 25,….)

Response: Corrected.

  1. Replace “ml” with “mL” (page 2, Scheme 1)

Response: Corrected.

  1. Avoid double repetition of stretching vibration value azomethine group ν(-HC=N-) (1630 cm-1, page 3, line 14). The value is clearly observed in Table 1 for both, CQ and HQ compounds.

Response: Corrected.

  1. Avoid double repetition of “…in in…” (page 3, line 17)

Response: Corrected.

  1. Replace “DMSO-d6” with “DMSO-d6” in all the manuscript (for example, page 3, line 1, below Table 1)

Response: Corrected.

  1. Please correct in all the manuscript the symbol for Celsius degree by using Symbol Insert function (°C) (for example, page 5, line 2, above Scheme 2)

Response: Corrected.  

  1. Replace “via” with “via”(page 12, paragraph 3.3)

Response: Corrected.

  1. Replace “Merk” with “Merck” (page 13, paragraph 3.3.3.)

Response: Corrected.

  1. How can you explain the satisfactory but not very high yields of the NQ/CQ/HQ complexes?

Response: NQ, CQ and HQ are produced from acid catalyzed condensation reaction which suffer from reverse reactions.

  1. Since the manuscript is entitled: “Metal complexes of Schiff bases…. structure and biological activity”, more details about the antiproliferative activity of the tested compounds should by presented in paragraph 3.5., even the figures 13S and 14S (from supplementary material) could be inserted here.

Response: Line 293: “From results in Table 7, it is clear that the presence of two phenolic groups (elec-tron-donating group) in HQ induces the activity more than electron withdrawing groups/atoms, namely nitro (NQ) or chloro (CQ). Relatively, nitro group is better than chlorine atom towards the activity. Phenolic and to less extent nitro group exhibits hydrogen bonding interaction as hydrogen bond acceptor. Phenolic compounds are known for their anti-inflammatory, antimicrobial, antioxidant and anticarcinogenic activity[42].”

Reviewer 2 Report

This manuscript by Mahmoud Sunjuk et al. describes the synthesis of Three Schiff bases of the quinoline-3-carbohydrazide core with three different aldehydes, and the formation of a variety of metal complexes, some structural studies and very preliminary biological evaluation.

In my view there is a serious lack of novelty in this manuscript and weak results to draw the readers' interest and attention.

The NMR study in paramagnetic complexes with no EPR studies is really too risky to present. 

Metal complexation did not give any additional biological benefit in the preliminary in vitro evaluation. 

In general many results/conclusions are drawn based on generalisations and extrapolations.

Some sections should be rephrased and improved.

Author Response

REVIEWER # 2:

In my view there is a serious lack of novelty in this manuscript and weak results to draw the readers' interest and attention.

Response: The originality points were added to the abstract, line 23: “The NQ preparation and the X-ray structure of NQ and CQ, as well as the transition metal complexes of NQ, CQ and HQ were reported for the first time”

Furthermore, the degree of originality was clarified by comparing the Schiff base complexes of the present work with those in the literature

lines 68: “In the literature, the azomethine nitrogen is conjugated with phenolic oxygen [14]–[18], quinoline [19]–[22] or phenolic oxygen and carbonyl [23], [24] to facilitate the chelating/coordinating properties of ligands to metal ions. The present work will describe the synthesis of Cu(II), Co(II), Ni(II), Cd(II), Cr(III) and Fe(III) complexes with three Schiff base ligands (NQ, CQ and HQ) which were prepared by condensation of Quinoline-3-carbohydrazide with aromatic aldehyde: 2-nitrobenzaldehyde, 2-chlorebenzaldehyde and 2,4-dihydroxybenzaldehyed, respectively (Scheme1). The ligands have an array of quinolone nitrogen, carbonyl oxygen, and azomethine nitrogen, as well as phenolic oxygen (in the case of HQ), rendering potential coordination sites to metal ions.”

The NMR study in paramagnetic complexes with no EPR studies is really too risky to present.

Response: We agree with the reviewer that is difficult to study the NMR of complexes since they are paramagnetic. However, some relatively sharp peaks were observed in the spectra of some complexes. This may be due to the fact that some protons are far from the metal center, and they are outside the paramagnetic sphere of the metal. There are some examples in the literature that report similar observations where some peaks appear in the range of 0–10 ppm for paramagnetic complexes [Pankratova, Y.; Aleshin, D.; Nikovskiy, I.; Novikov, V.; Nelyubina, Y. In Situ NMR Search for Spin-Crossover in Heteroleptic Cobalt(II) Complexes. Inorg. Chem. 2020, 59, 7700–7709; Ramadan, S.; Hambley, T.W.; Kennedy, B.J.; Lay, P.A. NMR Spectroscopic Characterization of Copper(II) and Zinc(II) Complexes of Indomethacin. Inorg. Chem. 2004, 43, 2943–2946; Nanthakumar, A.; Fox, S.; Murthy, N.N.; Karlin, K.D. Inferences from the 1H-NMR Spectroscopic Study of an Antiferromagnetically Coupled Heterobinuclear Fe(III)−(X)−Cu(II) S = 2 Spin System (X = O2−, OH−). J. Am. Chem. Soc. 1997, 119, 3898–3906].

Metal complexation did not give any additional biological benefit in the preliminary in vitro evaluation.

Response: Actually, the reviewer comment is true. There was no synergistic effect between the metal and ligand.

Line 35: “Nevertheless, metal ions did not exhibit significant synergistic effect on the antiproliferative activ-ity of the ligands investigated”

In general many results/conclusions are drawn based on generalisations and extrapolations.

Response: The discussion of FTIR (Lines 201-250) and NMR (Lines 256-269) was revised to become less general and more specific to the cases and compared with literature cases.

Comments on the Quality of English Language

Some sections should be rephrased and improved.

Response: Extensive editing of text was done.

Reviewer 3 Report

 In the paper, three  substituted quinoline-3-carbohydrazide ligands NQ, CQ and HQ are studied.

Single crystal structures of the first two ligands were obtained. The cif files show that the structures have been satisfactorily determined. However the cif files should contain the dimensions (bond lengths and angles) including the H atoms. In addition details of the hydrogen bonds should be included in both structures, not just for one of them.

Details of the two crystal structure determinations (cell dimensions, R values etc) should be given in a Table in the Supplementary Publication.  The experimental section in the text should include details of how the H atoms bonded to N and O were located and how they were refined. It is usual to include the donor – acceptor distance and the donor-H…accetpr angle iwhen discussing hydrogen bonds. These could be included in a Table in the Supplementary Publication.  One distance 2.051 needs to be truncated. It would be preferable if Figure 2 was rotated horizontally by 180 to give an orientation equivalent to that of Figure 1

Complexes of the ligands with Cu(II), Co(II), Ni(II), Cd(II), Cr(III), Fe(III) are then  investigated. It is surprising that no crystal structures of these complexes were obtained as that would be very helpful in discussing them. Maybe crystals could not be obtained as that sometimes happens. But in that case surely the paper requires a detailed discussion of what the structures might be which is surely relevant to the discussion of activity. Indeed there no figures suggesting what the structures might be. All that we are given is  some vague guesses as to what the structures might be which is unsatisfactory.  

The paper then includes spectroscopic properties which include NMR and FTIR spectra, magnetic susceptibility of a few of the complexes which are routine.

The antimicrobial activity of the ligands and their metal complexes are then reported.

 This work is routine and more work needs to be done on the metal complexes before it can be accepted

Author Response

REVIEWER #3:

Comments and Suggestions for Authors

 In the paper, three  substituted quinoline-3-carbohydrazide ligands NQ, CQ and HQ are studied.

Single crystal structures of the first two ligands were obtained. The cif files show that the structures have been satisfactorily determined. However the cif files should contain the dimensions (bond lengths and angles) including the H atoms. In addition details of the hydrogen bonds should be included in both structures, not just for one of them.

Response: The C-H and N-H bond lengths and angles are not included in the CIFs as all H atoms were included in calculated positions, so these dimensions merely reflect these constrained values. The H-atom coordinates are in the CIF which is accessible from the CCDC by those that are interested. H-bonds are discussed in both structures and the H-bond details are in the Supporting Information as requested (Table 4S and 8S).

Details of the two crystal structure determinations (cell dimensions, R values etc) should be given in a Table in the Supplementary Publication. 

Response: Tables 1S and 5S were added to provide the required information.

The experimental section in the text should include details of how the H atoms bonded to N and O were located and how they were refined.

Response: This has been done. Line 322-329: “X-ray crystallographic data were collected on an Oxford Diffraction Gemini CCD diffractometer with Cu-Kα radiation for CQ·H2O (1.54184 Ǻ) or Mo-Kα radiation for NQ·H2O (0.71073 Ǻ). The samples were cooled to 190 K with an Oxford Cryosystems Desktop Cooler. Data reduction and empirical absorption corrections were performed using CrysAlisPro (Rigaku Oxford Diffraction). The structures were solved by Direct Methods by SHELXT and refined with SHELXL [43]. All calculations were carried out within the WinGX package [44]. All non-H atoms were refined aniostropically. The data in CIF format have been deposited with the CCDC (numbers 2279657 and 2279658).”

It is usual to include the donor – acceptor distance and the donor-H…accetpr angle iwhen discussing hydrogen bonds. These could be included in a Table in the Supplementary Publication. 

Response: This has been done in Table 4S and 8S.

One distance 2.051 needs to be truncated. It would be preferable if Figure 2 was rotated horizontally by 180 to give an orientation equivalent to that of Figure 1

Response: The orientation of the structure does not affect its appearance and we prefer the figure that we have in the paper as it illustrates all of the features discussed in the text.

Complexes of the ligands with Cu(II), Co(II), Ni(II), Cd(II), Cr(III), Fe(III) are then  investigated. It is surprising that no crystal structures of these complexes were obtained as that would be very helpful in discussing them. Maybe crystals could not be obtained as that sometimes happens. But in that case surely the paper requires a detailed discussion of what the structures might be which is surely relevant to the discussion of activity. Indeed there no figures suggesting what the structures might be. All that we are given is  some vague guesses as to what the structures might be which is unsatisfactory. 

Response: crystals of complexes were not obtained. Scheme 2 gives the proposed structure of complexes. However, the coordination was dependent on the nature of ligand. Line 477-481: “As evidenced by FTIR spectroscopy, in the solid state, HQ is coordinated to the metal ion via azomethine nitrogen and phenolic oxygen. On the other hand, NQ and CQ are coordinated through azomethine nitrogen and carbonyl oxygen. 1H-NMR experiments indicated that the azomethine site in the ligands is the most preferable site for coordination to metal ion in the solution.”

The paper then includes spectroscopic properties which include NMR and FTIR spectra, magnetic susceptibility of a few of the complexes which are routine. The antimicrobial activity of the ligands and their metal complexes are then reported. This work is routine and more work needs to be done on the metal complexes before it can be accepted.

Response: The originality points were added to the abstract, line 23: “The NQ preparation and the X-ray structure of NQ and CQ, as well as the transition metal complexes of NQ, CQ and HQ were reported for the first time”

Furthermore, the degree of originality was clarified by comparing the Schiff base complexes of the present work with those in the literature

lines 68: “In the literature, the azomethine nitrogen is conjugated with phenolic oxygen [14]–[18], quinoline [19]–[22] or phenolic oxygen and carbonyl [23], [24] to facilitate the chelating/coordinating properties of ligands to metal ions. The present work will describe the synthesis of Cu(II), Co(II), Ni(II), Cd(II), Cr(III) and Fe(III) complexes with three Schiff base ligands (NQ, CQ and HQ) which were prepared by condensation of Quinoline-3-carbohydrazide with aromatic aldehyde: 2-nitrobenzaldehyde, 2-chlorebenzaldehyde and 2,4-dihydroxybenzaldehyed, respectively (Scheme1). The ligands have an array of quinolone nitrogen, carbonyl oxygen, and azomethine nitrogen, as well as phenolic oxygen (in the case of HQ), rendering potential coordination sites to metal ions.”

Round 2

Reviewer 2 Report

The authors addressed all concerns adequately.

Reviewer 3 Report

The authors have responded well to my comments and made appropriate changes. I consider that the paper is now of an acceptable standard